# Phage-displayed synthetic library and screening platform for nanobody discovery

**Baolong Xia[1†], Ah-Ram Kim[1†], Feimei Liu[2], Myeonghoon Han[1], Emily Stoneburner[1], Stephanie Makdissi[3], Francesca Di Cara[3], Stephanie E Mohr[1,4], Aaron Ring[2,5]\*, Norbert Perrimon[1,4,6]\***

[1]Department of Genetics, Blavatnik Institute, Harvard Medical School, Boston, United States; [2]Department of Immunobiology, Yale School of Medicine, Boston, United States; [3]Department of Microbiology and Immunology, Dalhousie University, Halifax, Canada; [4]Drosophila RNAi Screening Center, Harvard Medical School, Halifax, Canada; [5]Division of Translational Science and Therapeutics, Fred Hutchinson Cancer Center, Seattle, United States; [6]Howard Hughes Medical Institute, Boston, United States

**\*For correspondence:**
aaronring@fredhutch.org (AR);
perrimon@receptor.med.harvard.edu (NP)

[†]These authors contributed equally to this work

## eLife Assessment

This **important** study presents an alternative platform for nanobody discovery using phage-displayed synthetic libraries. The evidence supporting the platform, which is used to isolate and validate nanobodies targeting Drosophila secreted proteins, is **compelling**. By making the library openly accessible, this provides an excellent resource to the wider scientific community. The paper presents a detailed protocol for nanobody screening; as this protocol is refined and optimized over time, this will increase the success rate for discovering nanobodies with improved properties using this alternative platform.

**Abstract** Nanobodies, single-domain antibodies derived from camelid heavy-chain antibodies, are known for their high affinity, stability, and small size, which make them useful in biological research and therapeutic applications. However, traditional nanobody generation methods rely on camelid immunization, which can be costly and time-consuming, restricting their practical feasibility. In this study, we present a phage-displayed synthetic library for nanobody discovery. To validate this approach, we screened nanobodies targeting various *Drosophila* secreted proteins. The nanobodies identified were suitable for applications such as immunostaining and immunoblotting, supporting the phage-displayed synthetic library as a versatile platform for nanobody development. To address the challenge of limited accessibility to high-quality synthetic libraries, this library is openly available for non-profit use.

## Introduction

Nanobodies, single-domain antibody fragments derived from camelid heavy-chain antibodies, are valuable tools in biological research and therapeutic trials due to their small size, superior stability, and high specificity for target antigens (*Muyldermans, 2013*; *Könning et al., 2017*). These features make nanobodies ideal for applications beyond traditional in vitro assays, including live-cell imaging and functional studies. For example, when fused to fluorescence proteins or functional domains,

intracellularly expressed nanobodies enable real-time visualization and manipulation of proteins within living cells, providing a useful approach to study cellular dynamics in native environments (*Rothbauer et al., 2006*; *Caussinus et al., 2011*; *Harmansa and Affolter, 2018*; *Schnider et al., 2024*).

Traditional methods for nanobody generation, which rely on animal immunization, can be costly and time-consuming and are also limited by immunologic tolerance to conserved antigens (*Banik et al., 2023*). To overcome these challenges, in vitro technologies such as yeast, phage, and ribosome display have emerged as valuable alternatives, enabling the generation of synthetic nanobody libraries for direct antigen selection without animal immunization (*Moutel et al., 2016*; *Zimmermann et al., 2018*; *McMahon et al., 2018*).

The yeast-display platform, as demonstrated by Kruse and colleagues (*McMahon et al., 2018*), has proven exceptionally useful for screening nanobodies that recognize distinct conformational states of target proteins, such as G-protein-coupled receptors (GPCRs), facilitating structural studies and functional assays. While yeast display offers benefits such as robust expression and efficient cell-based screening, it also demands substantial antigen quantities and incurs high costs due to the need for magnetic- or fluorescence-activated cell sorting (MACS/FACS). These factors can limit the accessibility of yeast display for high-throughput or resource-limited applications. Moreover, due to limitations in yeast cell transformation efficiency, yeast display libraries have relatively low diversity ($10^8$ variants in the Kruse yeast library), potentially limiting the chances of identifying high-affinity nanobodies.

Phage display, meanwhile, is a well-established technology in antibody discovery (*Smith and Petrenko, 1997*; *Nagano and Tsutsumi, 2021*; *Hutchings and Sato, 2024*). Phage display generally requires less antigen, enables precise control over screening conditions (e.g. temperature, pH, chemical composition), and supports rapid, iterative rounds of selection. Given these benefits, we sought to create a phage-displayed synthetic nanobody library to leverage the diversity of a well-designed synthetic library with the added flexibility of phage display, providing an alternative platform for nanobody discovery.

Here, we describe the construction of a phage-displayed synthetic nanobody library using DNA templates originally used to create a yeast display library (*McMahon et al., 2018*). We validate the platform by screening for nanobodies against a panel of *Drosophila*-secreted proteins, with applications in immunostaining and immunoblotting analysis. This phage display-based platform offers a streamlined, cost-effective solution for nanobody discovery, broadening access to synthetic libraries for biological research and biomedical applications.

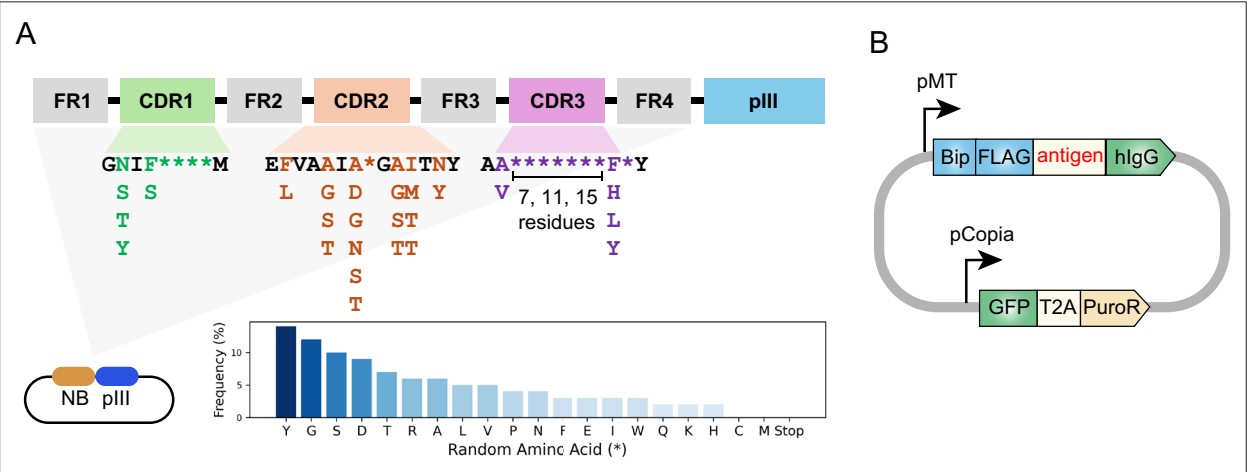

**Figure 1.** Design of synthetic nanobody library and antigen-expressing vector. (**A**) Schematic of synthetic nanobody library design. Nanobody sequences were inserted at the 5' end of pIII sequence in the phagemid. Nanobody sequences contain constant framework (FR1-FR4) and randomized regions within the CDRs. The randomization in highly variable positions is indicated as asterisks (*). The frequencies of amino acids in the variable positions are indicated in the bar chart. Cysteine and methionine were eliminated from design to avoid chemical reactivity. The CDR3 region was designed with varying lengths (7, 11, or 15 residues). (**B**) Schematic of antigen-expressing vector. Antigen expression was driven by an inducible metallothionein (MT) promoter. The antigen is fused to an N-terminal BiP signal peptide and FLAG-tag, as well as a C-terminal human Immunoglobulin G (hIgG) Fc domain. GFP-T2A-PuroR cassette is driven by Copia promoter for fluorescence labeling and antibiotics selection.

## Results

### Design and construction of phage-displayed synthetic nanobody library

To adapt the previously developed yeast-displayed nanobody library to a phage display platform, we used existing DNA templates originally reported by *McMahon et al., 2018* to construct a phage-displayed nanobody library. In this library, the nanobody sequences include constant framework and designed position-specific variations of three complementarity-determining regions (CDRs; *Figure 1A*). CDRs form the binding interface between nanobodies and antigens, and thus are crucial for nanobody diversity and specificity. The partial randomization in the residues flanking CDRs recapitulates the observed variations at these positions of 93 unique nanobodies deposited in the Protein Data Bank. The more thorough randomization in highly variable positions within CDRs reflects the high diversity of the library. Moreover, the CDR3 region was designed with varying lengths (7, 11, or 15 residues) to promote diverse binding modes and increase the potential for high-affinity binders, providing an additional dimension of diversity and complexity to the nanobody library.

M13 bacteriophage is a non-lytic filamentous bacteriophage and is commonly used for phage display. M13 bacteriophage contains single-stranded DNA genome packaged within a protein capsid formed by the major coat protein pVIII and the minor coat proteins pIII, pVI, pVII, and pIX (*Jia and Xiang, 2023*). To construct a phage-displayed nanobody library, the DNA sequences were inserted into 5' end of pIII sequence in a phagemid vector (*Figure 1A*). Next, the phagemid library was introduced into bacteria by massive electroporation. This resulted in a yield of $2.4 \times 10^{10}$ individual clones, suggesting a high level of diversity in the phage library. Following helper phage infection, the phages were packaged in the bacteria periplasm and secreted into the culture medium, producing a phage library with nanobodies displayed on the phage surface via fusion to the pIII coat protein.

### Purification of *Drosophila* secreted proteins as antigens for nanobody screening

Cell-cell communication is primarily mediated by secreted and transmembrane proteins, which is of significant research interest and has therapeutic potential. To address the challenge of producing antibodies against secreted and membrane proteins—especially the need for proper protein maturation and post-translational modifications—we used *Drosophila* S2 cells to produce 8 *Drosophila* secreted proteins as antigens for nanobody screening: Carbonic anhydrase-related protein B (CARPB), Myoinhibiting peptide precursor (Mip), Salivary gland-derived secreted factor (Sgsf), neuropeptide Nesfatin-1, Midkine and pleiotrophin 1 (Miple1), Amnesiac (Amn), CG9849, and CG13965.

The antigen expression vector was designed with two modules: one for antigen secretion and the other for stable cell line generation (*Figure 1B*). The antigen-expression module includes an inducible metallothionein promoter to prevent potential toxicity from constitutive expression, along with a BiP signal peptide for secretion, a FLAG-tag for detection, and a human Immunoglobulin G (hIgG) Fc domain for purification. The second module contains GFP-T2A-PuroR under constitutively active Copia promoter, enabling both GFP-based labeling and puromycin selection in serum-free ESF921 media.

Following transfection, puromycin selection was used to establish stable cell lines. Initial adherent cultures were adapted to suspension culture to increase cell density for efficient antigen production. Antigen expression was induced with $CuSO_4$, after which conditioned media was harvested and the proteins were purified from conditioned medium using Protein A resin.

### Nanobody screening with phage-displayed nanobody library

To screen for nanobodies that bind these antigens, we first coated 96-well MaxiSorp plates with mCherry-hIgG for negative selection and antigen-hIgG for positive selection (*Figure 2A*). Then, the phage-displayed nanobody library was incubated with a mCherry-hIgG-coated plate to remove phages with affinity to the plate or the hIgG domain. Next, unbound phages were transferred to antigen-hIgG-coated plates to select phages with affinity to the antigen. After stringent washing, the phages that remained bound to the antigen-coated plate were eluted by infecting bacteria in an 'elution by infection' manner. The infected bacteria were used to amplify the selected phages in the presence of helper phages to generate phages used in subsequent iterative selection steps. To enrich for nanobodies with high affinity, in the next rounds of selection, we reduced the antigen amount

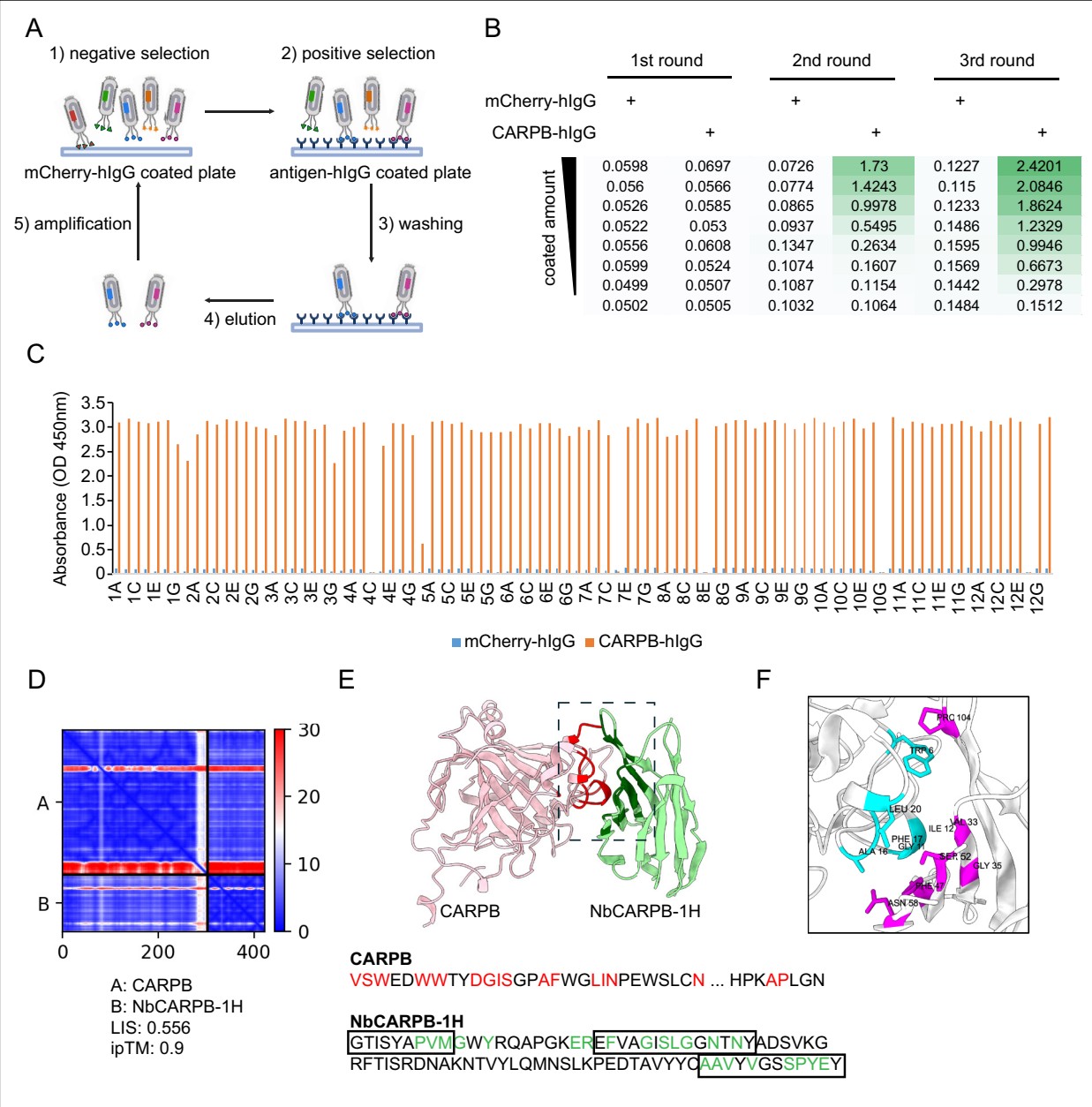

**Figure 2.** Nanobody screen for CARPB. (**A**) Schematic of nanobody screening. The mCherry-hIgG-coated plate was used for negative selection and the antigen-hIgG-coated plate was used for positive selection. Three iterative rounds of screening were performed for each antigen. (**B**) ELISA with polyclonal phages from three rounds of selection in CARPB screen campaign. The mCherry-hIgG and CARPB-hIgG were coated in decreasing amounts across different rows. The intensity of green color represents the ELISA signal strength. (**C**) ELISA with monoclonal phages from the third round of selection in the CARPB screen campaign. 96 monoclonal phages were tested against mCherry-hIgG and CARPB-hIgG. (**D**) Predicted Aligned Error (PAE) map generated using AlphaFold-Multimer for the CARPB-NbCARPB-1H complex. Low PAE values, shown in blue, indicate accurate predictions for the interaction interfaces, depicted in the top right and bottom left quadrants. High Local Interaction Score (LIS) and interface TM-score (ipTM) values confirm the structural reliability of the interface. (**E**) Predicted 3D structure of the CARPB-NbCARPB-1H complex, showing the interaction interface. The region enclosed by the dotted box corresponds to the interaction interface visualized in panel F. Residues involved in the interaction (within 8 Å distance) are marked in red (CARPB) and green (NbCARPB-1H) in the structure and protein sequences. CDRs in the protein sequence are marked by boxes. The interaction predominantly occurs within these CDR regions. (**F**) Neighboring residues between CARPB and NbCARPB-1H, showing residues within a 5 Å distance. Cyan indicates residues of CARPB, and magenta indicates residues of NbCARPB-1H. Each residue is labeled with its residue name and number.

The online version of this article includes the following figure supplement(s) for figure 2:

**Figure supplement 1.** Nanobody screen for Nesfatin-1.

*Figure 2 continued on next page*

**Figure supplement 2.** Nanobody screen for Mip.

**Figure supplement 4.** Nanobody screen for Amn.

**Figure supplement 5.** Failed nanobody screens for Sgsf, Miple1, and CG13965.

coated on the plate and increased the number of washing cycles to gradually increase the selection stringency (see Materials and methods).

After three rounds of selection, we evaluated the selection outcomes with an enzyme-linked immunosorbent assay (ELISA) using the polyclonal phages after each round of selection. Typically, the ELISA signal increased after 2–3 rounds of selection, indicating that phages with affinity for the antigen were progressively enriched in the polyclonal population (*Figure 2B*). For the CARPB and Nesfatin-1 screen campaigns (*Figure 2B*, *Figure 2—figure supplement 1A*), we observed specific ELISA signals against antigen-hIgG but not against mCherry-hIgG even when testing the polyclonal phages, suggesting that the majority of the phage population in these samples specifically recognizes the antigens.

Next, we identified positive hits from the nanobody screens by ELISA using monoclonal phages after the third round of selection. The monoclonal phages exhibiting specific ELISA signals against antigen-hIgG rather than mCherry-hIgG were defined as positive hits for each screen campaign. As expected, most monoclonal phages exhibited specific antigen recognition in the CARPB and Nesfatin-1 screen campaigns (*Figure 2C*, *Figure 2—figure supplement 1B*), consistent with the polyclonal phage ELISA results. The nanobody sequences of these hits were identified by Sanger sequencing of the phagemids. In total, we identified 5 candidate nanobodies for CARPB (*Figure 2*), 7 candidate nanobodies for Nesfatin-1 (*Figure 2—figure supplement 1*), 2 candidate nanobodies for Mip (*Figure 2—figure supplement 2*), 1 candidate nanobody for CG9849 (*Figure 2—figure supplement 3*) and 2 candidate nanobodies for Amn (*Figure 2—figure supplement 4*), demonstrating that the phage-displayed nanobody library is a versatile platform for discovering nanobodies against various antigens. These nanobody sequences were listed in *Table 1*. However, we failed to identify nanobodies against Sgsf, Miple1, and CG13965 (*Figure 2—figure supplement 5*), suggesting room for further improvement of our current library and screening platform (see Discussion).

To predict the structural basis of antigen-nanobody interactions, we used AlphaFold-Multimer (*Evans et al., 2022*) to model CARPB-nanobody complexes and evaluated the interaction using both interface pTM (ipTM) score (*Evans et al., 2022*) and Local Interaction Score (LIS) (*Kim et al., 2024*). The ipTM score reflects the structural accuracy of the predicted complex, while LIS indicates the potential interaction strength. Among the five candidate nanobodies for CARPB, one nanobody, NbCARPB-1H, exhibited high LIS and ipTM scores, indicating a reliable interaction model (*Figure 2D*). The Predicted Aligned Error (PAE) map further supported the accuracy of the complex prediction, showing low PAE values (shown as blue color area) in the interaction interface, indicating a reliable structure prediction (*Figure 2D*). The predicted 3D structure revealed specific interactions between the complementarity-determining regions (CDRs) of NbCARPB-1H and residues in both the N-terminal (CARPB$^{24-52}$) and C-terminal (CARPB$^{276-277}$) regions of CARPB (*Figure 2E and F*). The successful structural modeling of NbCARPB-1H highlights its reliable interaction with CARPB, validating the accuracy of AlphaFold-Multimer predictions for antigen-nanobody complexes.

## Nanobodies recognize membrane-tethered forms of the antigens

To further validate that the identified nanobodies can recognize the antigens, we tethered each antigen to the cell surface by glycosylphosphatidylinositol (GPI) anchoring or mCD8 fusion. Four of the five antigens (CARPB, Mip, Nesfatin-1, and CG9849, but not Amn) were successfully presented on cell surface as detected by N-terminal FLAG-tag immunostaining. To test nanobody binding, the antigen-expressing cells were incubated with nanobody-displaying phages, followed by detection of phage binding on cell surface. In contrast to what we observed for an irrelevant nanobody Nb.b201, a nanobody against human serum albumin (*McMahon et al., 2018*), the positive nanobody-displaying phages could recognize the antigens on the cell surface (*Figure 3A*).

Since phage display potentially results in multivalent nanobody presentation, we next asked if the monovalent form of the nanobodies still recognizes the antigens. To do this, we incubated the

**Table 1.** Protein sequences of the nanobodies identified in this study.

| nanobody | sequence |
| --- | --- |
| NbCARPB-1H | QVQLQESGGGLVQAGGSLRLSCAASGTISYAPVMG WYRQAPGKEREFVAGISLGGNTNYADSVKGRFTISR DNAKNTVYLQMNSLKPEDTAVYYCAAVYVGSSPYEYWGQGTQVTVSS |
| NbCARPB-2A | QVQLQESGGGLVQAGGSLRLSCAASGNISHYSIMGW YRQAPGKEREFVAAINIGATTNYADSVKGRFTISRDNAK NTVYLQMNSLKPEDTAVYYCAAYAARRPGYEYWGQGTQVTVSS |
| NbCARPB-2E | QVQLQESGGGLVQAGGSLRLSCAASGTISAHSVMGWY RQAPGKEREFVAGIAYGGNTNYADSVKGRFTISRDNAKNT VYLQMNSLKPEDTAVYYCAVASHYTRAPATAHDYWGQGTQVTVSS |
| NbCARPB-4D | QVQLQESGGGLVQAGGSLRLSCAASGYISQPGYMGWYRQ APGKERELVAAITSGGITYYADSVKGRFTISRDNAKNTVYLQM NSLKPEDTAVYYCAVSYASSYLYSYWGQGTQVTVSS |
| NbCARPB-5F | QVQLQESGGGLVQAGGSLRLSCAASGSIFIGFMGWYRQAP GKERELIAGITSGGSTYYADSVKGRFTISRDNAKNTVYLQMN SLKPEDTAVYYCAVVYGGYAYWPAWHDYWGQGTQVTVSS |
| NbMip-4G | QVQLQESGGGLVQAGGSLRLSCAASGNIFNRGAMGWYRQAP GKERELVAAINQGTNTYYADSVKGRFTISRDNAKNTVYLQMNS LKPEDTAVYYCAVHPTYKSHLGYWGQGTQVTVSS |
| NbMip-4H | QVQLQESGGGLVQAGGSLRLSCAASGTIFFRGVMGWYRQA PGKERELVAAISRGANTYYADSVKGRFTISRDNAKNTVYLQMN SLKPEDTAVYYCAVYFPNEGGHYYWGQGTQVTVSS |
| NbNesfatin1-1A | QVQLQESGGGLVQAGGSLRLSCAASGTISTFTFMGWYRQ APGKEREFVAAIGYGGITNYADSVKGRFTISRDNAKNTVYL QMNSLKPEDTAVYYCAAPYGYSYPQFYAFKYWGQGTQVTVSS |
| NbNesfatin1-1E | QVQLQESGGGLVQAGGSLRLSCAASGNIFEPASMGWYRQ APGKERELVAAINRGAITYYADSVKGRFTISRDNAKNTVYLQM NSLKPEDTAVYYCAASRYYGAIFLYWGQGTQVTVSS |
| NbNesfatin1-1G | QVQLQESGGGLVQAGGSLRLSCAASGNIFRYRAMGWYRQ APGKERELVAAIAAGGTTYYADSVKGRFTISRDNAKNTVYLQ MNSLKPEDTAVYYCAVGRYIVSVYRDDYWYWGQGTQVTVSS |
| NbNesfatin1-1H | QVQLQESGGGLVQAGGSLRLSCAASGSIFYLVDMGWYRQAP GKEREFVATIATGGITYYADSVKGRFTISRDNAKNTVYLQMNS LKPEDTAVYYCAARVYDGTSDWRHYYYWGQGTQVTVSS |
| NbNesfatin1-2A | QVQLQESGGGLVQAGGSLRLSCAASGNISPYAAMGWYRQA PGKEREFVAAIARGSTTYYADSVKGRFTISRDNAKNTVYLQM NSLKPEDTAVYYCAVRYLGYTVKGILHIYWGQGTQVTVSS |
| NbNesfatin1-2C | QVQLQESGGGLVQAGGSLRLSCAASGYIFYRYTMGWYRQA PGKERELVAGINRGGITNYADSVKGRFTISRDNAKNTVYLQM NSLKPEDTAVYYCAVIARAQWYLAYWGQGTQVTVSS |
| NbNesfatin1-3C | QVQLQESGGGLVQAGGSLRLSCAASGSIFRFYGMGWYRQ APGKEREFVAAIAYGTTTNYADSVKGRFTISRDNAKNTVYLQ MNSLKPEDTAVYYCAVRGYNTADSVRRYDYWGQGTQVTVSS |
| NbCG9849-15G | QVQLQESGGGLVQAGGSLRLSCAASGSIFPWRGMGWYRQ APGKERELVAAISGGANTNYADSVKGRFTISRDNAKNTVYLQ MNSLKPEDTAVYYCAAGTLTAGQHRYWGQGTQVTVSS |
| NbAmn-4D | QVQLQESGGGLVQAGGSLRLSCAASGSIFLIFAMGWYRQA PGKEREFVATIGYGATTNYADSVKGRFTISRDNAKNTVYLQ MNSLKPEDTAVYYCAVRGFYLRNPPDGYNIRHRYWGQGTQVTVSS |
| NbAmn-10A | QVQLQESGGGLVQAGGSLRLSCAASGTIFLGIYMGWYRQA PGKERELVAAIALGASTYYADSVKGRFTISRDNAKNTVYLQM NSLKPEDTAVYYCASRDAADIYDHDFWYWGQGTQVTVSS |

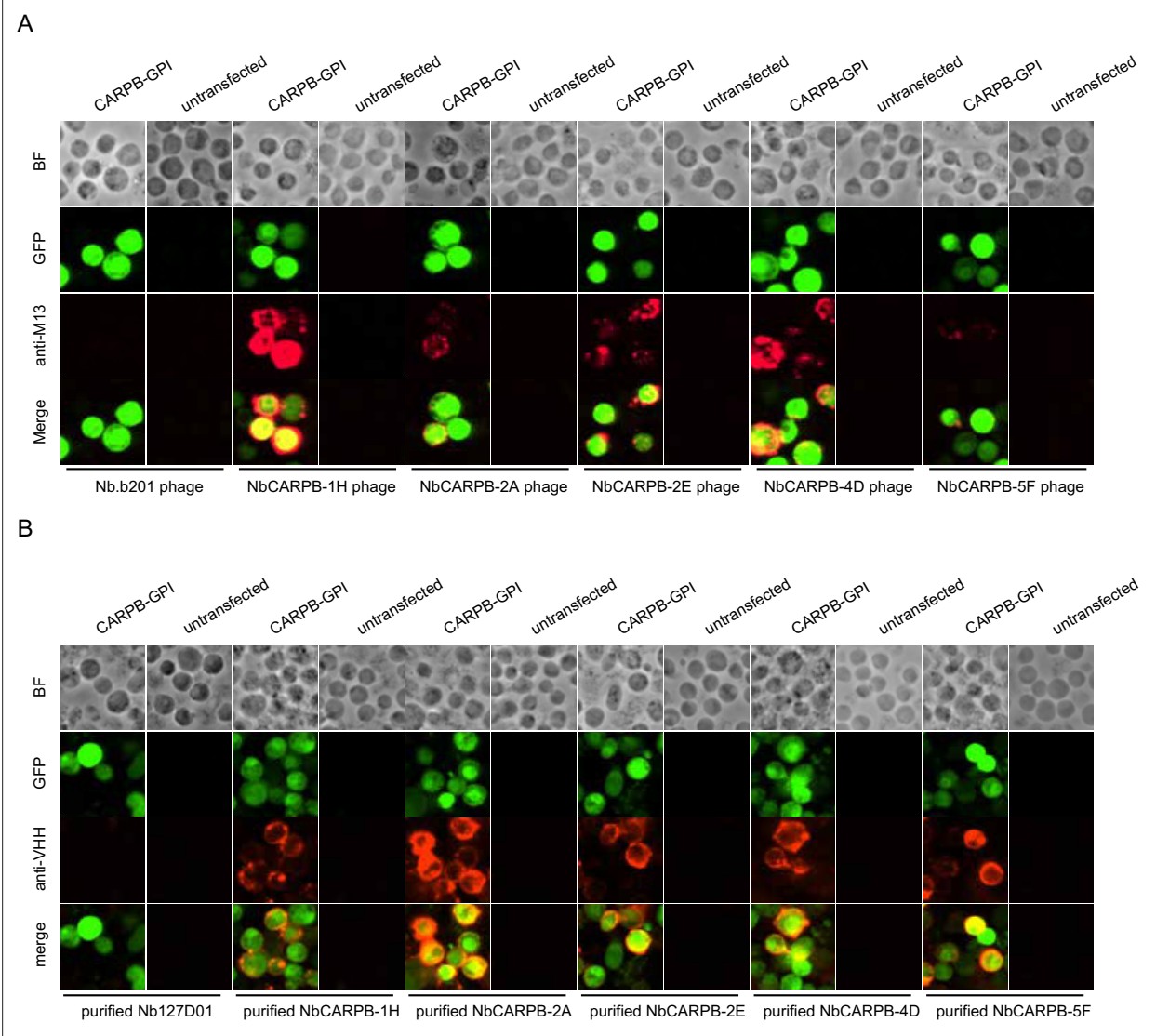

**Figure 3.** Immunostaining of membrane-tethered CARPB with nanobodies. (**A**) CARPB-expressing cells stained with nanobody-displaying phages. GFP signals mark cells expressing membrane-tethered CARPB. Nb.b201 phage, an irrelevant nanobody displaying phage, was used as negative control. BF, bright field. (**B**) CARPB-expressing cells stained with purified nanobodies. GFP signals label the cells expressing membrane-tethered CARPB. Nb127D01 is an irrelevant nanobody used as a negative control. BF, bright field.

antigen-expressing cells with purified nanobodies, and we observed similar results (*Figure 3B*), further suggesting that these nanobodies specifically recognize their target proteins.

## Nanobody application in immunostaining and immunoblotting

To test downstream applications of the nanobodies, NbMip-4G, identified in our nanobody screening against Mip (*Figure 2—figure supplement 2*), was evaluated for immunostaining and immunoblotting applications. Mip is a neuropeptide that regulates sleep homeostasis, food intake, and mating behavior (*Oh et al., 2014*; *Min et al., 2016*; *Jang et al., 2017*). We observed a strong cytoplasmic immunostaining signal in cells with smaller nuclei in the *Drosophila* adult gut (*Figure 4A*), consistent with a previous observation based on single-cell RNA-seq data that *Mip* is expressed in the enteroendocrine cells (*Guo et al., 2019*). Importantly, the immunostaining signal disappeared when *Mip* expression was knocked down by RNA interference (RNAi) (*Mex >Mip RNAi[v106076]*), further supporting the specificity of the NbMip-4G.

Immunoblotting analysis of adult fly hemolymph with NbMip-4G also demonstrated its utility in protein detection. A clear reduction in Mip protein levels was observed in *Mip* mutant (*Mip[1]*) or RNAi

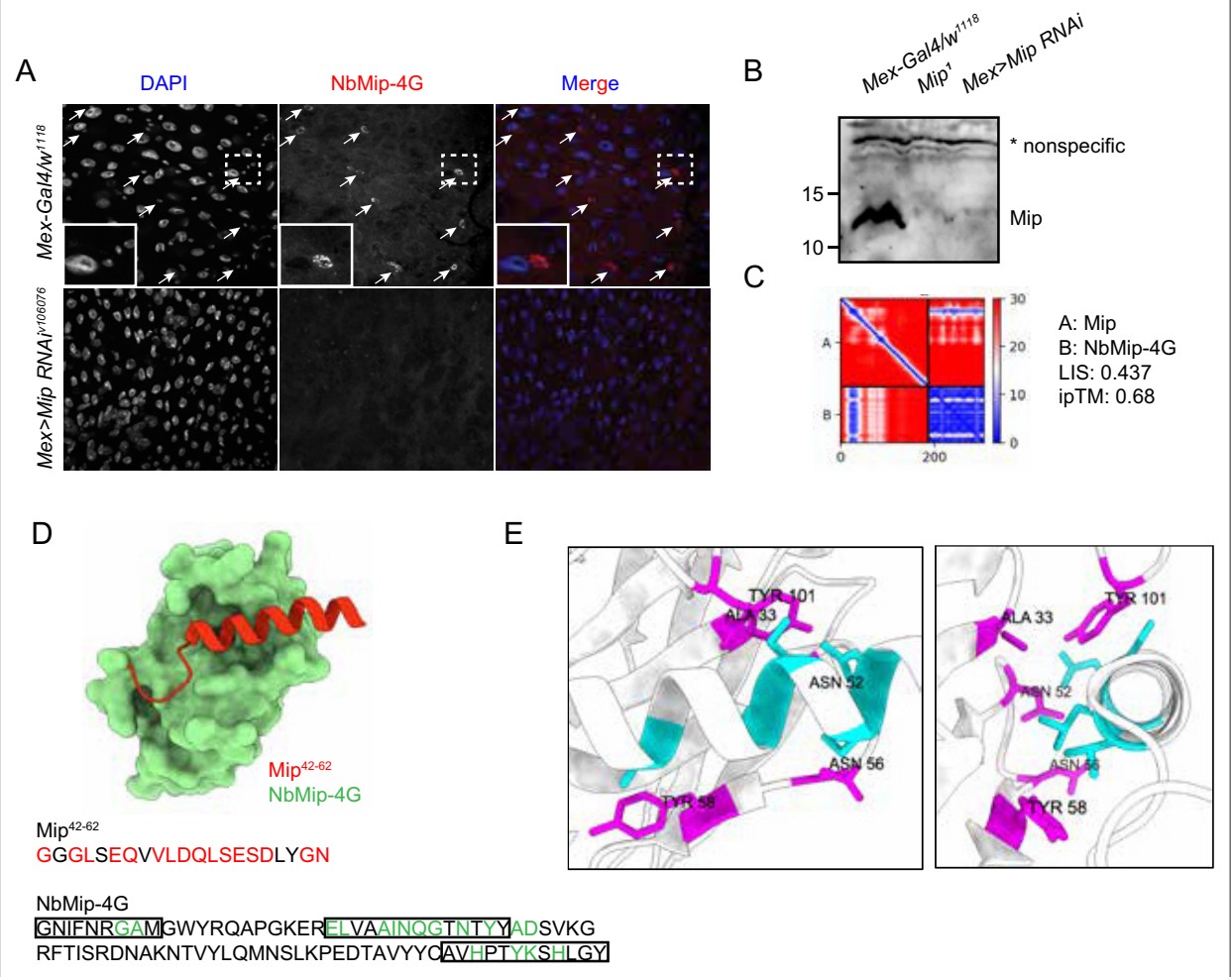

**Figure 4.** Immunostaining and immunoblotting with NbMip-4G nanobody. (**A**) Immunostaining of *Drosophila* adult intestine with NbMip-4G nanobody. (**B**) Immunoblotting detection of Mip in *Drosophila* adult hemolymph with NbMip-4G nanobody. A nonspecific band is included as a loading control. (**C**) Predicted Aligned Error (PAE) map generated using AlphaFold-Multimer for the Mip-NbMip-4G complex. Low PAE values, shown in blue, indicate accurate predictions for the interaction interfaces, depicted in the top right and bottom left quadrants. High LIS and ipTM values confirm the structural reliability of the interface. (**D**) Predicted 3D structure of the Mip-NbMip-4G complex. The surface representation shows NbMip-4G in green and the interacting helical segment of Mip (residues 42–62) in red. Key residues involved in the interaction are marked in red (Mip) and green (NbMip-4G) in the protein sequences. CDRs in the protein sequence are marked by boxes. The interaction predominantly occurs within these CDR regions. (**E**) Neighboring residues between Mip and NbMip-4G, showing residues within a 5 Å distance. Cyan indicates residues of Mip, and magenta indicates residues of NbMip-4G. Each residue in NbMip-4G is labeled with its residue name and number.

knockdown (*Mex >Mip RNAi[v106076]*) samples compared to control (*Mex-Gal4/w[1118]*) (*Figure 4B*). These results confirm that NbMip-4G can specifically detect Mip in immunoblotting applications.

To further explore the molecular interaction between Mip and NbMip-4G, we predicted their complex using AlphaFold-Multimer and assessed the structural reliability of the interaction. The PAE map (*Figure 4C*) indicated low PAE values at the interaction interface between Mip and NbMip-4G, confirming the accuracy of the structural prediction. The predicted 3D structure of the Mip-NbMip-4G complex revealed that NbMip-4G primarily interacts with a helical segment of Mip spanning residues 42–62 (*Figure 4D*). Interacting residues, predominantly located within the CDRs of NbMip-4G, were identified through contact interface analysis (*Figure 4D and E*). These structural insights elucidate the molecular basis of the high specificity and binding affinity of NbMip-4G for Mip, reinforcing its potential as a versatile tool for detecting and studying Mip in functional studies.

## Discussion

In this study, we constructed a phage-displayed nanobody library and established a versatile platform for rapid nanobody discovery. Furthermore, we successfully identified nanobodies against various *Drosophila* secreted proteins using this platform. Our current phage display library has higher diversity compared with yeast display library ($10^{10}$ vs $10^8$), potentially increasing the likelihood of identifying high-affinity binders. Additionally, whereas the yeast screening platform requires expensive and tedious MACS/FACS for each round of selection, the phage screening platform has much simpler selection steps and a faster turnaround time, making the phage screening platform a streamlined, cost-effective solution for nanobody discovery.

The Kruse nanobody library design, which served as the foundation for our work, incorporates optimized randomization within complementarity-determining regions (CDRs) while maintaining a robust framework (*McMahon et al., 2018*). We further adapted this design for phage display, enabling greater scalability and access. Unlike many existing libraries that are commercially restricted or require expensive licensing, we have democratized access to our platform by making it openly available for non-profit use to address the field's broader challenge of limited accessibility to high-quality synthetic libraries.

Despite its advantages, our library and approach still face challenges. For example, we screened nanobodies against eight *Drosophila* secreted proteins and successfully identified nanobodies for five of them. For the remaining three antigens, we did not observe any monoclonal phage with specific affinity to the antigens after 3 or 4 rounds of selection. Further increasing library diversity, optimizing the screening strategy, and/or improving antigen quality might increase the success rate. Moreover, for these hits we identified, some nanobodies (e.g. Nesfatin-1 nanobodies in *Figure 2—figure supplement 1*) only showed binding affinity in ELISA and failed to recognize membrane-tethered forms of the antigens, possibly due to low affinity. Further affinity maturation by random mutagenesis or directed evolution might increase affinity of these antibodies to the antigens.

Overall, we show that our phage-displayed nanobody library provides a highly efficient, cost-effective platform for nanobody discovery. By making this resource openly accessible to the academic and non-profit research community, we hope to foster innovation and collaboration across the scientific community, overcoming financial and technical barriers that have limited the use of synthetic libraries. Our library represents a step forward in addressing the diverse and complex needs of nanobody discovery, offering a valuable alternative to other existing platforms.

## Materials and methods

### Key resources table

| Reagent type (species) or resource | Designation | Source or reference | Identifiers | Additional information |
|---|---|---|---|---|
| Gene (*Drosophila melanogaster*) | CARPB | FlyBase | FLYB: FBgn0052698 | |
| Gene (*Drosophila melanogaster*) | Mip | FlyBase | FLYB: FBgn0036713 | |
| Gene (*Drosophila melanogaster*) | Sgsf | FlyBase | FLYB: FBgn0266261 | |
| Gene (*Drosophila melanogaster*) | Nesfatin-1 | FlyBase | FLYB: FBgn0052190 | |
| Gene (*Drosophila melanogaster*) | Miple1 | FlyBase | FLYB: FBgn0027111 | |
| Gene (*Drosophila melanogaster*) | Amn | FlyBase | FLYB: FBgn0086782 | |
| Gene (*Drosophila melanogaster*) | CG9849 | FlyBase | FLYB: FBgn0034803 | |
| Gene (*Drosophila melanogaster*) | CG13965 | FlyBase | FLYB: FBgn0032834 | |

*Continued on next page*

*Continued*

| Reagent type (species) or resource | Designation | Source or reference | Identifiers | Additional information |
|---|---|---|---|---|
| Cell line (*D. melanogaster*) | S2 | DRSC | FLYB: FBbt:00005737 | |
| Cell line (*D. melanogaster*) | S2R+ | DRSC | FLYB: FBtc0000150 | |
| Strain, strain background (*Escherichia coli*) | SS320 (MC1061 F') Electrocompetent Cells | Biosearch | 60512–2 | For phage library construction |
| Strain, strain background (*Escherichia coli*) | One Shot TOP10 Chemically Competent *E. coli* | Invitrogen | C404010 | |
| Strain, strain background (*Escherichia coli*) | One Shot OmniMAX 2 T1$^R$ *E. coli* | Invitrogen | C854003 | |
| Strain, strain background (*Escherichia coli*) | One Shot ccdB Survival 2 T1$^R$ Competent Cells | Invitrogen | A10460 | |
| Strain, strain background (*Escherichia coli*) | BL21 Competent *E. coli* | NEB | C2530H | |
| Antibody | M13 Major Coat Protein Antibody (RL-ph1) HRP | Santa Cruz biotechnology | sc-53004 HRP | 1:1000 dilution for ELISA |
| Antibody | M13 Major Coat Protein Antibody (RL-ph1) Alexa Fluor 647 | Santa Cruz biotechnology | sc-53004 AF647 | 1:500 dilution for immunostaining |
| Antibody | Alexa Fluor 647 AffiniPure Goat Anti-Alpaca IgG, VHH domain | Jackson ImmunoResearch | 128-605-230 | 1:500 dilution for immunostaining |
| Antibody | Peroxidase AffiniPure Goat Anti-Alpaca IgG, VHH domain | Jackson ImmunoResearch | 128-035-230 | 1:5000 dilution for immunoblotting |
| Commercial assay or kit | Effectene Transfection Reagent | Qiagen | 301425 | |

## Construction of phage-displayed nanobody library

Library construction followed an established protocol (*McMahon et al., 2018*). Nanobody library DNA fragments were digested with the restriction enzymes FseI and NsiI at 37 °C for 1 hr and purified using gel extraction (QIAGEN). Similarly, the phagemid vector p443 was linearized by FseI and NsiI digestion and purified by gel extraction.

The purified nanobody DNA fragments and linearized vector were ligated overnight at 16 °C using T4 DNA ligase (NEB). The ligation products were cleaned using a QIAquick PCR Purification Kit (QIAGEN) and eluted in water. The purified phagemid constructs were then electroporated into electrocompetent *E. coli* strain SS320 (Biosearch) at 2,500 V, 125 Ω, and 50 μF. Immediately following electroporation, cells were recovered in 5 mL of S.O.C medium (Corning) by shaking at 37°C and 220 rpm for 1 hr. M13K07 helper phage particles were added at a final concentration of $10^{10}$ phage/mL and incubated for 1 hr at 37 °C without antibiotics. Next, the culture was expanded in 1 L of 2YT medium supplemented with Tetracycline (at final concentration of 10 μg/mL), kanamycin (50 μg/mL), and carbenicillin (100 μg/mL), followed by overnight incubation in a 37 °C shaking incubator.

The overnight culture was centrifuged, and phage particles were precipitated from the supernatant by adding PEG/NaCl to a final volume ratio of 1:5 and incubating on ice for 30 min. The precipitated phage was resuspended in PBS, supplemented with 10% glycerol, thoroughly mixed, aliquoted,

and stored at –80 °C. The constructed library resulted in $2.4\times10^{10}$ transformants, with each aliquot containing $10^{13}$ phage particles.

## Sequences of the *Drosophila* secreted proteins

The protein sequence of Nesfatin-1 was adapted from previous literature (*Yeom et al., 2021*). Putative mature protein sequences for seven *Drosophila* secreted proteins (CARPB, Mip, Sgsf, Miple1, Amn, CG9849, and CG13965) assessed in this study were determined based on the predicted sequences excluding signal peptide or transmembrane domain from the full sequences (https://www.flyrnai.org/tools/nanobody/web/protein). The protein sequences of the eight *Drosophila* secreted proteins are as follows.

### CARPB (FBgn0052698)
VSWEDWWTYDGISGPAFWGLINPEWSLCNKGRRQSPVNLEPQRLLFDPNLRPMHIDKHRISGLITNTG
HSVIFTAGNDTVANYDGMQTPVNISGGPLSYRYRFHEIHMHYGLNDQFGSEHSVEGYTFPAEIQIFGY
NSQLYANFSDALNRAQGIVGVSILLQLGDLSNAELRMLTDQLERIRYGGDEAFVKRLSIRGLLPDTDHYMTY
DGSTTAPACHETVTWVVLNKPIYITKQQLHALRRLMQGSPDHPKAPLGNNYRPPQPLLHRPIRTNIDFKTTK
SNGKAACPTMYREVYYKATSWKQN.

### Mip (FBgn0036713)
NLVASGSAGSPPSNEPGGGGLSEQVVLDQLSESDLYGNNKRAWQSLQSSWGKRSSSGDVSDPDIYMTG
HFVPLVITDGTNTIDWDTFERLASGQSAQQQQQQPLQQQSQSGEDFDDLAGEPDVEKRAWKSMN
VAWGKRRQAQGWNKFRGAWGKREPTWNNLKGMWGKRDQWQKLHGGWGKRSQLPSN.

### Sgsf (FBgn0266261)
YRIIESNEVPKTCPALNKDIIFEEPHLNNNQREFYDVREIPRRLHFNSNKKEIQSENWLLRIIRIKTINDGPRHKV
TEIKEDKHMDGFAKRLFNILKKTLKMHEPSYKHNNDQKDLFVLKKNHFPPEHHIVKREPVHYNYM.

### Miple1 (FBgn0027111)
STVLGTTEGQETPLALPVAEQTQPTTAIQGEVWEEDDHEVLIRNERGTKSDGLSCRYGKNPWTECDTK
TNTRSRTLTLKKGDPACDQTRTIQKKCKKACRYEKGSWSECATGQMTRADKLKASSDPSCEATRVIKK
NCKPGKSKDKSAKEQRKNKDKAARKGRV.

### Amn (FBgn0086782)
LRRRVVSGSKGSAALALCRQFEQLSASRRERAEECRTTQLRYHYHRNGAQSRSLCAAVLCCKRSYIPRPNFS
CFSLVFPVGQRFAAARTRFGPTLVASWPLCNDSETKVLTKWPSCSLIGRRSVPRGQPKFSRENPRALSPSLL
GEMR.

### Nesfatin-1 (Precursor protein: NUCB1, FBgn0052190)
LPVTQNKKDHKEAAESSTPATADVETALEYERYLREVVEALEADPEFRKKLDKAPEADIRSGKIAQELDYVN
HHVRTKLDEI.

### CG9849 (FBgn0034803)
STTISIPITTQDIIAGDVFFEILSPSELEYTYRLRPAKDFGSAFSERLEGVPLVITDPPGACQEIRNARDLNGGVA
LIDRGECSFLTKTLRAEEAAGALAAIITEYNPSSPEFEHYIEMIHDNSQQDANIPAGFLLGKNGVIIRSTLQR
LKRVHALINIPVNLTFTPPSKINHPPWLGW.

### CG13965 (FBgn0032834)
QETPAAESSPASPTDGETSPVTEASSIGELTQTTEAGSEVTESPTNSTDMVNSTDNPDPNGSPDPENG
GDPFVKPGSHIKGPRHVRAHDGFHSLKTEKHWASWNDAFTTPRP.

## Purification of secreted antigens using stable *Drosophila* S2 cell lines

*Drosophila* S2 cells cultured in ESF921 medium (Expression Systems) were transfected with an antigen expression vector as described (*Kim et al., 2022*). Following transfection, puromycin selection was

used to establish stable cell lines. Adherent cultures were gradually adapted to suspension conditions to enhance cell density and antigen production. Antigen expression was induced by adding 500 µM CuSO$_4$ to suspension cultures. Conditioned medium (CM) containing secreted antigens was harvested 4–5 days post-induction. CM was then clarified by centrifugation at 1000×$g$ for 10 min and filtered through a 0.22 µm filter to remove residual debris.

Protein A resin (1 mL, GenScript) was prepared by washing with 25 mL TBS in a gravity purification column. CM containing antigen-hIgG (50–250 mL) was passed through the resin by gravity flow, and the flow-through was collected. The resin was then washed three times with 25 mL TBS to remove unbound material. To elute antigen-hIgG, the resin was treated with 5 mL of 0.1 M glycine (pH 3.5), and the eluate was collected into a 15 mL conical tube preloaded with 0.5 mL Tris-HCl (pH 8.5), equivalent to 10% of the elution buffer volume. An additional 2 mL of TBS was passed through the resin, and the eluate was collected into the same tube. The combined eluate was mixed thoroughly.

The pooled eluate was concentrated using an Amicon Ultra-15 centrifugal filter (50 kDa cutoff; Millipore Sigma) at 4150×$g$ and 4 °C for 30 min, with TBS washes repeated three times. The protein concentration of the final sample was measured at A280 using a NanoDrop spectrophotometer, with absorbance values adjusted based on the protein extinction coefficients.

## Nanobody screening by phage display

Purified antigen-hIgG proteins were diluted in PBS (1 µg/well for the first round, 0.5 µg/well for the second round and 0.25 µg/well for the third round) and coated onto Nunc MaxiSorp 96-well plates (Thermo Fisher Scientific, 442404) at 4 °C overnight. Antigen-coated plates were blocked with freshly prepared 5% skim milk in PBST (MPBST; PBS with 0.05% Tween-20) at room temperature (RT) for 1–2 hr. For negative selection, MaxiSorp plates were coated with mCherry-hIgG (1 µg/well), blocked similarly with 5% MPBST, and processed alongside the antigen plates.

For the first round of screening, nanobody-displaying phage libraries were thawed (1.5 mL tube) and diluted in 4.5 mL MPBST. A 60 µL aliquot of the diluted phages was added to each well of the mCherry-hIgG-coated plates at RT for 1 hr to remove nonspecific binders. The unbound phage-containing solution was then transferred to antigen-hIgG-coated plates. To minimize phage loss during transfer, the negative selection plate wells were washed with 90 µL MPBST, and the wash solution was also transferred to the antigen-coated plates, bringing the final volume to 150 µL. Antigen plates were incubated with the phage solution at RT for 1 hr and washed extensively with TBST (10 times in the first round, 20 times in the second round, and 30 times in the third round).

To recover antigen-bound phages, freshly prepared OmniMAX *E. coli* cells (Invitrogen, C854003) were grown to an OD600 of ~0.6 and added to each well of the washed antigen plates (50 µL per well). The plates were incubated at RT for 30 min with gentle shaking. Following incubation, the bacteria were collected and mixed with helper phages at a multiplicity of infection (MOI) of ~20. The mixture was incubated at 37 °C for 30 min with shaking and subsequently transferred to 36 mL of 2xYT medium supplemented with ampicillin (100 µg/mL) and kanamycin (50 µg/mL). The cultures were incubated overnight at 37 °C with shaking at 220 rpm to propagate the phages.

Phages were precipitated from the overnight supernatant using PEG/NaCl (20% PEG8000, 2.5 M NaCl; 9 mL per 36 mL supernatant), incubated at 4 °C for 30 min, and centrifuged at 12,000×$g$ for 30 min at 4 °C. The phage pellet was resuspended in 2 mL PBS, centrifuged at 12,000×$g$ for 5 min to remove debris, and 1 mL was used for subsequent screening rounds. Phages obtained after the final round of screening were used for downstream applications, including polyclonal and monoclonal phage ELISA, and for monoclonal phage production.

## Polyclonal phage ELISA

Purified antigen-hIgG and mCherry-hIgG proteins were serially diluted, coated onto MaxiSorp plates, and blocked as described above. Polyclonal phage libraries from the 1st, 2nd, and 3rd selection rounds were incubated with the antigen-coated plates at room temperature (RT) for 1 hr. After incubation, the plates were washed three times with TBST and then incubated with an anti-M13-HRP secondary antibody (Santa Cruz biotechnology, sc-53004 HRP) in blocking solution at RT for 1 hr.

Following incubation, the plates were washed again with TBST three times. 100 µL TMB substrate solution (Thermo Fisher Scientific, 34028) was added to each well, and color development was

monitored. The reaction was stopped by adding 50 µL 0.4 M $H_2SO_4$ per well, and absorbance was measured at 450 nm (OD450) to quantify ELISA signals.

## Monoclonal phage ELISA

To isolate individual nanobody-displaying phages, 96 phage-producing colonies were picked from the 3rd round of selection and inoculated into 96-well deep plates containing 100 µL 2xYT medium supplemented with carbenicillin. The plates were incubated at 37 °C for 6–8 hr until OD600 reached approximately 0.6. Helper phages were added to the cultures and incubated at RT for 30 min. The culture was then supplemented with 1 mL per well of 2xYT containing both carbenicillin and kanamycin and incubated overnight at 37 °C with shaking.

The culture supernatants containing monoclonal phages were collected by centrifuging the plates. These supernatants were used as the primary antibody for the monoclonal phage ELISA. To prepare antigen-coated plates, purified antigen-hIgG and mCherry-hIgG proteins (0.5 µg/mL) were coated onto MaxiSorp plates and blocked as described above. A mixture of the phage-containing supernatant and blocking solution (1:1) was added to the antigen-coated plates and incubated at RT for 1 hr.

After washing with TBST for 10 times, the plates were incubated with an anti-M13-HRP secondary antibody in blocking solution at RT for 1 hr. Following incubation, the plates were washed again with TBST three times. 100 µL TMB substrate solution (Thermo Fisher Scientific, 34028) was added to each well, and color development was monitored. The reaction was stopped by adding 50 µL 0.4 M $H_2SO_4$, and absorbance was measured at 450 nm (OD450) to quantify the ELISA signals.

## Cloning, expression, and purification of nanobody-ALFA-His in BL21 competent cells

To generate a cloning vector for bacterial nanobody expression with an ALFA-tag for detection and a His-tag for purification, a pET-26b-Nb-GGA plasmid compatible with BsaI-mediated Golden Gate Assembly (GGA) was constructed. A DNA fragment containing _CmR-ccdB_, the ALFA-tag coding sequence, and the His-tag coding sequence was synthesized with a point mutation in the _ccdB_ gene to eliminate the BsaI recognition site. The synthetic DNA fragment was inserted into NcoI/XhoI-linearized pET-26b vector by HiFi cloning (NEB). The phagemids of the positive nanobodies identified in the monoclonal ELISA step were used to amplify PCR products with nanobody CDS flanked by BsaI-GGA cloning sites. The resulting PCR products were utilized to clone pET-26b-Nanobody-ALFA-His in One Shot ccdB Survival 2 T1R Competent Cells (Invitrogen, A10460).

The production and purification of nanobody-ALFA-His followed previously published protocols (**_Kim et al., 2022_**). Briefly, BL21 competent cells (NEB) were transformed with the pET-26b-Nanobody-ALFA-His plasmid and plated on LB plate supplemented with kanamycin (50 µg/mL). After overnight incubation at 37 °C, a single colony was inoculated into 10 mL LB medium supplemented with kanamycin and grown at 37 °C in a shaking incubator overnight.

The overnight culture was used to inoculate 50 mL LB medium with kanamycin in a 250 mL Erlenmeyer flask. The culture was incubated at 37 °C with shaking until the OD600 reached 0.6–0.8, at which point IPTG was added at a final concentration of 0.3 mM to induce protein expression. The culture was incubated overnight at 15 °C with shaking.

After overnight induction, bacterial cells were harvested by centrifugation at 6000×_g_ for 15 min at 4 °C. The pellet was subjected to periplasmic preparation using B-PER II reagent (Thermo Fisher Scientific, 78260); details described in **_Kim et al., 2022_**. The periplasmic fraction filtered by 0.22 µm filter was supplemented with 10 mM imidazole and used for purification using $Ni^{2+}$ affinity chromatography (Cytiva, 17531801). The periplasmic sample was passed through a gravity-flow column containing 1 mL of $Ni^{2+}$ resin pre-washed with TBS with 10 mM imidazole. The column was washed with 20 mL of TBS with 20 mM imidazole (3 times), and nanobody-ALFA-His was eluted in 1 mL fractions using TBS with increasing imidazole concentrations (100 mM, 250 mM, 500 mM). Eluates were analyzed by SDS-PAGE, and fractions containing purified nanobody were pooled, dialyzed, and stored at −80 °C.

## Live cell staining with nanobody-displaying phages or purified nanobodies

S2R + cells were transfected with antigen-GPI or antigen-mCD8 plasmids for transfection validation. After 2 days of transfection, cells were harvested and washed with PBS. The cells were incubated with a

blocking solution (e.g. 5% skim milk in PBS) at 4 °C for 30 min, followed by incubation with nanobody-displaying phages or purified nanobodies in the blocking solution for 1 hr at 4 °C. After three washes with PBS, cells were incubated with secondary antibodies anti-M13-647 (Santa Cruz Biotechnology, sc-53004 AF647) or anti-VHH-647 (Jackson ImmunoResearch, 128-605-230) for 30 min at 4 °C. After three additional washes with PBS, the cells were imaged using an IN Cell high-throughput confocal microscopy system (MicRoN Core, Harvard Medical School).

## Immunostaining of adult fly intestine

Flies were maintained on standard medium at 25 °C. *Mip* knockdown was achieved using a *Mip* RNAi line (VDRC v106076) under the control of a *Mex-Gal4* driver (BDSC 91367) for 20 days at 25 °C. Flies (*w¹¹¹⁸* crossed with *Mex-Gal4*) were used as controls. Adult flies were dissected in 1×PBS to collect intestinal tissues, which were fixed in 4% methanol-free formaldehyde in PBS at RT for 1 hr.

Following fixation, tissues were permeabilized with 0.1% Triton X-100 in PBS (PBST), with three solution changes every 10 min for a total of 30 min. Tissues were then blocked in 5% normal goat serum (NGS) in PBST at RT for 1 hr. Samples were incubated overnight at 4 °C with primary antibody NbMip-4G (1:10 dilution) diluted in 5% NGS in PBST.

After incubation, tissues were washed three times with PBST (15 min per wash) and then incubated with secondary antibody anti-VHH-647 (Jackson ImmunoResearch, 128-605-230, 1:500 dilution) in 5% NGS in PBST at RT for 2 hr. Following secondary antibody incubation, tissues were washed four times in 0.05% Tween-20 in PBS (15 min per wash) and given a final wash in 1×PBS. The tissues were mounted in ProLong Gold Antifade Mounting Medium (Invitrogen) and imaged using a confocal microscope.

## Immunoblotting of adult fly hemolymph

Hemolymph samples were collected from control flies (*w¹¹¹⁸* crossed with *Mex-Gal4*), *Mip* mutant flies (*Mip¹*, a kind gift from Jongkyeong Chung, Seoul National University), and *Mip*-RNAi flies (VDRC v106076 crossed with *Mex-Gal4*) using a previously reported protocol (*Pinelli et al., 2024*). The protein samples were subject to SDS-PAGE (12% gel). After transfer, the membrane was blocked with 5% BSA/TBST at 4 °C overnight. The membrane was incubated with the primary antibody NbMip-4G (1:300) at RT for 1 hr.

The membrane was then washed three times with TBST (15 min per wash) before incubation with secondary antibody Goat Anti-Alpaca IgG, VHH domain (Jackson ImmunoResearch 128-035-230, 1:5000). The membrane was washed three times (15 min per wash) in TBST, developed using Clarity Western ECL Substrate (Bio-Rad), and imaged using a ChemiDOC Imaging System (Bio-Rad).

## Antigen-nanobody complex prediction using AlphaFold-Multimer

For the prediction of antigen-nanobody complexes using AlphaFold-Multimer (AFM), we employed LocalColabFold version 1.5.5 (*Mirdita et al., 2022*). It integrates AFM version 2.3.2 (*Evans et al., 2022*) and generates multiple sequence alignments via MMseqs2 (version 14-7e284). Computations were performed on the Harvard O2 high-performance computing cluster. The ipTM score was used to evaluate the overall structural accuracy and confidence of the entire protein complex (*Evans et al., 2022*). While ipTM provides a global metric, it may overlook localized interactions in small or flexible regions. To address this limitation, we calculated the Local Interaction Score (LIS), which leverages low Predicted Aligned Error (PAE) regions to pinpoint interacting domains, ignoring high PAE regions likely associated with non-interacting areas (*Kim et al., 2024*). The contact interface was identified using two criteria: residues with a Predicted Aligned Error (PAE) less than 12 and an inter-residue distance of less than 8 Å between C-beta atoms (or C-alpha atoms for glycine residues). The code for LIS analysis is available in https://github.com/flyark/AFM-LIS (*Kim, 2025*). For the visualization of the predicted structures, ChimeraX was utilized (*Meng et al., 2023*).

## Declaration of generative AI used in the writing process

During the preparation of this manuscript, the authors used OpenAI's ChatGPT to improve the readability and language of the text. All intellectual content, scientific accuracy, and conclusions were reviewed, verified, and edited by the authors, who take full responsibility for the final version of the manuscript.

## Acknowledgements

We thank Dr. Andrew C Kruse for the nanobody library oligos and suggestions. We are grateful to the Research Computing Group at Harvard Medical School for access to the O2 High Performance Compute Cluster. This work was supported by NIH NIGMS P41 GM132087 (NP) and NIH 5R24OD035556 (NP). AK was supported by the Postdoctoral Fellowship Program (Nurturing Next-generation Researchers) through the National Research Foundation of Korea (NRF) funded by the Ministry of Education (2021R1A6A3A14039622). NP is an investigator of Howard Hughes Medical Institute. This article is subject to HHMI's Open Access to Publications policy. HHMI lab heads have previously granted a non-exclusive CC BY 4.0 license to the public and a sublicensable license to HHMI in their research articles. Pursuant to those licenses, the author-accepted manuscript of this article can be made freely available under a CC BY 4.0 license immediately upon publication.

## Additional information

### Funding

| Funder | Grant reference number | Author |
| --- | --- | --- |
| National Institutes of Health | NIH NIGMS P41 GM132087 | Norbert Perrimon |
| National Institutes of Health | NIH 5R24OD035556 | Norbert Perrimon |
| National Research Foundation | 2021R1A6A3A14039622 | Ah-Ram Kim |

The funders had no role in study design, data collection and interpretation, or the decision to submit the work for publication.

### Author contributions

Baolong Xia, Conceptualization, Data curation, Formal analysis, Validation, Investigation, Methodology, Writing - original draft, Project administration, Writing - review and editing; Ah-Ram Kim, Conceptualization, Formal analysis, Validation, Investigation, Visualization, Writing - original draft; Feimei Liu, Resources, Formal analysis, Investigation; Myeonghoon Han, Resources, Investigation; Emily Stoneburner, Investigation; Stephanie Makdissi, Francesca Di Cara, Validation, Investigation; Stephanie E Mohr, Funding acquisition; Aaron Ring, Conceptualization, Resources, Supervision; Norbert Perrimon, Conceptualization, Supervision, Writing - original draft, Project administration

### Author ORCIDs
Baolong Xia http://orcid.org/0000-0003-2536-0267
Ah-Ram Kim https://orcid.org/0000-0001-9597-6759
Stephanie E Mohr https://orcid.org/0000-0001-9639-7708
Norbert Perrimon https://orcid.org/0000-0001-7542-472X

Reviewer #2 (Public review): https://doi.org/10.7554/eLife.105887.3.sa1
Author response https://doi.org/10.7554/eLife.105887.3.sa2

## Additional files

### Supplementary files
MDAR checklist

### Data availability
The code for LIS analysis is available in https://github.com/flyark/AFM-LIS (*Kim, 2025*).

The following dataset was generated:

| Author(s) | Year | Dataset title | Dataset URL | Database and Identifier |
|---|---|---|---|---|
| Xia B, Kim A-R, Liu F, Han M, Stoneburner E, Makdissi S, Di Cara F, Mohr SE, Ring AM, Perrimon N | 2025 | Phage-displayed synthetic library and screening platform for nanobody discovery-source data | https://doi.org/10.5061/dryad.f4qrfj77j | Dryad Digital Repository, 10.5061/dryad.f4qrfj77j |

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
