## [Editor Report · eLife Assessment]

This **important** study presents an alternative platform for nanobody discovery using phage-displayed synthetic libraries. The evidence supporting the platform, which is used to isolate and validate nanobodies targeting Drosophila secreted proteins, is **compelling**. By making the library openly accessible, this provides an excellent resource to the wider scientific community. The paper presents a detailed protocol for nanobody screening; as this protocol is refined and optimized over time, this will increase the success rate for discovering nanobodies with improved properties using this alternative platform.

---

## [Referee Report · Reviewer #2 (Public review)]

Summary:

In this study, authors propose an alternative platform for nanobody discovery using a phage-displayed synthetic library. Authors relied on DNA templates originally created by McMahon et al. (2018) to build the yeast-displayed synthetic library. To validate their platform, authors screened for nanobodies against 8 Drosophila secreted proteins. Nanobody screening has been performed with phage-displayed nanobody libraries followed by an enzyme-linked immunosorbent assay (ELISA) to validate positive hits. Nanobodies with higher affinity have been then tested for immunostaining and immunoblotting applications using Drosophila adult guts and hemolymph, respectively.

Strengths:

The authors presented a detailed protocol with various and complementary approaches to select nanobodies and test their application for immunostaining and immunoblotting experiments. Data are convincing and the manuscript is well-written, clear and easy to read.

Weaknesses:

When using membrane-tethered forms of the antigens to test the affinity of nanobodies identified by ELISA, many nanobodies fail to recognize the antigens. While authors suggested a low affinity of these nanobodies for their antigens, this hypothesis has not been tested in the manuscript.

Improving the protocol at each step for nanobody selection would greatly increase a successful rate for nanobodies discovery with high affinity.

---

## [Author Response]

The following is the authors’ response to the original reviews.

**Reviewer #1 (Public review):**
Summary:Using highly specific antibody reagents for biological research is of prime importance. In the past few years, novel approaches have been proposed to gain easier access to such reagents. This manuscript describes an important step forward toward the rapid and widespread isolation of antibody reagents. Via the refinement and improvement of previous approaches, the Perrimon lab describes a novel phage-displayed synthetic library for nanobody isolation. They used the library to isolate nanobodies targeting Drosophila secreted proteins. They used these nanobodies in immunostainings and immunoblottings, as well as in tissue immunostainings and live cell assays (by tethering the antigens on the cell surface).Since the library is made freely available, it will contribute to gaining access to better research reagents for non-profit use, an important step towards the democratisation of science.Strengths:(1) New design for a phage-displayed library of high content.(2) Isolation of valuble novel tools.(3) Detailed description of the methods such that they can be used by many other labs.

We are grateful for these supportive comments.

Weaknesses:My comments largely concentrate on the representation of the data in the different Figures.

We have made adjustments according to the reviewer’s recommendations.

**Reviewer #2 (Public review):**
Summary:In this study, the authors propose an alternative platform for nanobody discovery using a phage-displayed synthetic library. The authors relied on DNA templates originally created by McMahon et al. (2018) to build the yeast-displayed synthetic library. To validate their platform, the authors screened for nanobodies against 8 Drosophila secreted proteins. Nanobody screening has been performed with phage-displayed nanobody libraries followed by an enzyme-linked immunosorbent assay (ELISA) to validate positive hits. Nanobodies with higher affinity have been tested for immunostaining and immunoblotting applications using Drosophila adult guts and hemolymph, respectively.Strengths:The authors presented a detailed protocol with various and complementary approaches to select nanobodies and test their application for immunostaining and immunoblotting experiments. Data are convincing and the manuscript is well-written, clear, and easy to read.

We thank the reviewer for these supportive comments.

Weaknesses:On the eight Drosophila secreted proteins selected to screen for nanobodies, the authors failed to identify nanobodies for three of them. While the authors mentioned potential improvements of the protocol in the discussion, none of them have been tested in this manuscript.

We prepared all eight antigens by single-step IgG purification (see Materials and Methods) without additional biophysical quality control (e.g., size-exclusion chromatography). Consequently, we cannot definitively determine whether the three “no-binder” cases resulted from the aggregation or misfolding of the antigens, versus gaps in our naive library’s sequence space. While approaches such as additional purification steps or affinity maturation of weak binders would likely rescue these difficult targets, comprehensive pipeline optimization is beyond the scope of establishing and validating the phage-displayed nanobody platform. We have clarified this limitation and suggested these strategies in third paragraph of the Discussion.

The same comment applies to the experiments using membrane-tethered forms of the antigens to test the affinity of nanobodies identified by ELISA. Many nanobodies fail to recognize the antigens. While authors suggested a low affinity of these nanobodies for their antigens, this hypothesis has not been tested in the manuscript.

We observed that several nanobodies with strong ELISA signals showed reduced binding to membrane-displayed antigens. This discrepancy may result from low affinity of the nanobodies or differences in post-translational modifications (e.g., glycosylation) and antigen context between secreted IgG-fusion proteins (used for panning/ELISA) and GPI- or mCD8-anchored proteins. In an ongoing work, we have performed affinity maturation of the nanobodies and successfully increased the affinity toward the target antigen. These results will be reported separately.

Improving the protocol at each step for nanobody selection would greatly increase the success rate for the discovery of nanobodies with high affinity.

We fully agree that systematic optimization—from antigen preparation (e.g., additionalpurification steps) through screening conditions (e.g., buffer composition, additional affinity-maturation steps)—could substantially increase the success rate and nanobody affinity. These represent important directions for future work.

**Recommendations for the authors:**

**Reviewer #1 (Recommendations for the authors):**
(1) Figure 3. The merge of two GFP channels does not make much sense. Can the authors not use artificial colours? And show the panels at higher resolution, such that a viewer can really see and judge what they are seeing? The same comments apply to all Supplementary Figures.

We appreciate the reviewer’s comment. In the revised Figure 3, we have replaced the cyan/green overlay with red/green overlay and used enlarged pictures so that GFP-positive cells and corresponding nanobody staining are clearly visible. We applied the same layout to all relevant Supplementary Figures.

(2) Figure 4. Also, in this Figure, it is not really possible to see what the authors say one should see. The resolution should be higher, and arrows or arrowheads should point to important structures.

We appreciate the reviewer’s comment. In the revised Figure 4A, we have added arrows to point to the immunostaining signal in cells with smaller nuclei and added inset panels to show a closer view of representative NbMip-4G staining.

**Reviewer #2 (Recommendations for the authors):**
(1) Images are sometimes quite small and difficult to interpret. For example, Figures S2C-D.

We thank the reviewer for this suggestion. In the revised figures, we have replaced the cyan/green overlay with red/green overlay and used enlarged pictures that clearly show GFP-positive cells alongside their corresponding nanobody staining.

(2) Supplemental figures are not always cited in the text.

Thank you for the comment. To eliminate this misunderstanding, we have updated the Nesfatin1 nanobody screen data as Supplementary Figure 1 and Mip nanobody screen data as Supplementary Figure 2. We have made the corresponding changes in the Results section.